# Learning from Observational Outcomes: Toward Causally-Aligned Language Model Fine-Tuning

## Abstract

Large language models are being widely used across industries to generate text that contributes directly to key performance metrics, such as medication adherence in patient messaging and conversion rates in content generation. Pretrained models, however, often fall short when it comes to aligning with human preferences or optimizing for business objectives. As a result, fine-tuning with good-quality labeled data is essential to guide models to generate content that achieves better results. Controlled experiments, like A/B tests, can provide such data, but they are often expensive and come with significant engineering, logistical, and ethical challenges. Meanwhile, companies have access to a vast amount of historical (observational) data that remains underutilized. In this work, we study the challenges and opportunities of fine-tuning LLMs using observational data. We show that while observational outcomes can provide valuable supervision, directly fine-tuning models on such data can lead them to learn spurious correlations. We present empirical evidence of this issue using various real-world datasets and propose DECONFOUNDLM, a method that explicitly removes the effect of known confounders from reward signals. In simulation experiments, DECONFOUNDLM more accurately recovers causal relationships and mitigates failure modes of methods that assume counterfactual invariance, achieving over 16% higher objective score than ODIN and other baselines, when entangled confounding is present.

## 1 Introduction

Large language models (LLMs) can be powerful tools for creating content that affects user behavior and supports business goals. From enhancing user engagement to increasing purchase likelihood, companies often aim to generate content that delivers measurable outcomes. Prior research has shown that pretrained LLMs can perform well in certain tasks, such as generating creative product ideas (Castelo et al., 2024) and predicting the likelihood of purchases (Arora et al., 2025). However, these models often struggle to capture human preferences and directly optimize for business outcomes (Goli & Singh, 2024; Ye et al., 2024), emphasizing the need for fine-tuning with labeled data to causally guide the models towards desired business outcomes. Yet obtaining the right kind of labeled data to support this alignment is difficult; human-labeled data and surveys can introduce bias due to artificial contexts (Yeh et al., 2024), and randomized experiments are often infeasible due to logistical and opportunity costs (Quin et al., 2024). This paper explores how to bridge this gap using an abundant but underutilized source of supervision available to firms: historical observational data.

Consider a news website that aims to improve the click-through rates (CTR) of news headlines. While they may not have the capacity to run controlled experiments, they may track how users respond to different headlines over time, which could be used for fine-tuning. However, directly fine-tuning on this data could be challenging because external factors such as time trends may influence both the content and the outcome. Furthermore, while running controlled experiments may already be impractical for a news website, the difficulty is amplified in healthcare settings, where fairness and ethical concerns further restrict experimentation and increase the importance of leveraging available historical data. In this paper, we examine both the opportunities and risks associated with using observational data, and we propose a novel method that corrects for confounding effects in the fine-tuning process.

Fine-tuned LLMs have been used in many domains from electronic health records (Wu et al., 2024) to astronomical data (Wang et al., 2024b) and social-science corpora; however most work does not discuss causal challenges or the pitfalls of learning from historical data. In business settings, methods such as Ye et al. (2024) and Angelopoulos et al. (2024) rely on experimental supervision, leaving the potential of firms' extensive historical logs underexplored. On the causal side, computer science work has documented biases in preference data used for fine-tuning (e.g., length bias and sycophancy) and proposed methods, such as ODIN (Chen et al., 2024), Wang et al. (2025), and Srivastava et al. (2025) that rely on the counterfactual invariance assumption. This assumption forces the reward estimates not to change when confounding attributes are changed. While counterfactual invariance may be plausible for human-rated LLM outputs, it is often violated in real-world applications where different variables can affect outcomes. In contrast, we propose a method to use abundant observational data and correct for observed confounders without assuming counterfactual invariance.

Our results in this paper show:

- **Risks of observational signals.** Using *StackExchange*, we show that naive fine-tuning on historical interactions can lead models to internalize spurious correlations.

- **Value of observational signals.** Using *Upworthy*, we show that observational data can provide valuable signals. We also highlight the role of regularization to suppress the confounding effects when using historical data.

- **Confounder correction.** We propose DECONFOUNDLM, a fine-tuning method that removes the influence of observed confounders from the reward signal. Our results show that this approach consistently improves model behavior, enabling it to focus on causally relevant attributes rather than superficial artifacts.

## 2 RELATED WORK

Our research spans three domains: (1) LLMs in science and business, (2) causal inference in econometrics and machine learning, and (3) alignment of LLMs with reward modeling under confounding. We summarize key contributions in each domain and highlight how our work extends current boundaries, particularly in aligning LLMs using observational data subject to confounding.

### 2.1 LLMS IN SCIENCE AND BUSINESS APPLICATIONS

LLMs have advanced applications across domains, from health records (Wu et al., 2024) and astronomy (Wang et al., 2024b) to social sciences. In business, most work leverages experimental supervision: Ye et al. (2024) use adaptive experiments for headline CTR, while Angelopoulos et al. (2024) fine-tune on A/B outcomes. Other efforts explore knowledge transfer (Wang et al., 2024a) and demand prediction (Lee, 2024). Together, these works highlight the promise of LLMs in business settings. However, aside from the last study, which focuses on demand prediction rather than content generation, these methods primarily rely on supervision signals obtained from controlled experiments, which are costly and limited in scope (Feit & Berman, 2019; Miller & Hosanagar, 2020), or on synthetic feedback that may reflect the biases of the teacher model. In contrast, our work investigates how to fine-tune LLMs using abundant observational data, while explicitly addressing the confounding factors that can mislead model learning.

### 2.2 CAUSAL INFERENCE WITH MACHINE LEARNING AND ECONOMETRICS

Econometric methods offer tools for causal learning under confounding. Chernozhukov et al. (2018) introduce Double Machine Learning (DML), using orthogonalized moment conditions and cross-fitting to reduce bias from regularization in high-dimensional settings. Farrell et al. (2021) extend this framework with a theoretical analysis for deep neural networks in semiparametric models. However, both methods assume exogeneity, limiting their utility when confounding is endogenous. IV-based approaches (Dikkala et al., 2020; Bennett et al., 2019; Singh & Zheng, 2023) handle endogeneity using adversarial or nonparametric estimation. Our work contributes to this line of research by adapting IV-based ideas for generative modeling with LLMs. Instead of estimating structural

parameters, our goal is to deconfound reward signals used in LLM fine-tuning, ensuring the model aligns with causal rather than spurious objectives.

### 2.3 LLM ALIGNMENT AND REWARD MODELING

RLHF has become standard for alignment (Ouyang et al., 2022), with PPO and DPO leveraging human preference data (Rafailov et al., 2024b; Zheng et al., 2023). Yet reward models inherit bias (Ntoutsi et al., 2020) and overfit to artifacts like length or sycophancy (Tien et al., 2022; Denison et al., 2024). To address these issues, several works have developed causal reward modeling frameworks. However, these methods typically assume counterfactual invariance, that the reward should not change when confounding attributes are perturbed—an assumption that often fails in practice. ODIN (Chen et al., 2024) removes known confounders (e.g., length) from the learned reward. Wang et al. (2025) train rewards to satisfy counterfactual invariance by construction. More recently, Srivastava et al. (2025) proposes a robust reward modeling method that enforces counterfactual invariance via counterfactual data augmentation. While the counterfactual invariance assumption may hold for LLM-generated answers evaluated by humans, it does not hold in many real-world applications, which we discuss more below.

$$f(\boldsymbol{F}, \boldsymbol{C}) = f(\boldsymbol{F}),$$

with $\boldsymbol{F}$ denoting meaningful features and $\boldsymbol{C}$ confounders. Approaches like ODIN (Chen et al., 2024), invariant training (Wang et al., 2025), and augmentation-based methods (Srivastava et al., 2025) rely on this assumption. However, it is overly restrictive in real-world business contexts where confounders (e.g., price, seasonality) legitimately affect outcomes. Our method, DECONFOUNDLM, relaxes invariance by estimating and removing spurious influences, enabling alignment with true causal drivers.

## 3 OBSERVATIONAL DATA: PITFALLS AND POTENTIAL

In this section, we examine both the pitfalls and potential of learning from historical observational data, relying on experiments using StackExchange and Upworthy data.

### 3.1 PITFALL: INTERNALIZING SPURIOUS CORRELATION

We illustrate how confounding in historical data can mis-specify rewards when fine-tuning LMs. Using Academia Stack Exchange, we mimic Askell et al. (2021) by treating user scores as preferences. In this dataset, engagement varies by weekday, specifically, we see a higher activity earlier in the week (Figure 1). Because of this pattern, scores partly reflect exposure rather than quality. To make this spurious signal explicit, we prepend a "Happy Monday!" marker to Monday answers, then construct preference pairs where the higher-scored answer is treated as preferred. Evaluated on 3,000 held-out questions, models trained on these preferences learn the weekday cue: compared to SFT, DPO amplifies the artifact, generating "Happy" 21.2% vs. 13.7% and "Monday" 11.8% vs. 9.1% of the time (both increases statistically significant). Full description and details of this experiment are presented in Appendix B.

### 3.2 POTENTIAL

In the previous section, we showed that historical data can induce spurious correlations. Here we ask: *What is its potential value?* When a firm lacks experimental data, can logs of content and observed performance still improve future predictions? This cannot be answered in purely observational settings because outcomes are not causally attributable to the content. We therefore use the Upworthy dataset Matias et al. (2021), which provides CTRs from controlled A/B tests. To simulate observational access, we retain a single headline package and its CTR from each test and discard the alternative; dataset statistics and preprocessing are discussed in Appendix C.

The realized CTR of a headline package depends not only on its intrinsic appeal but also on exogenous factors such as audience composition and temporal context. For example, surges of sports-related traffic during major events or elevated engagement with political content during election

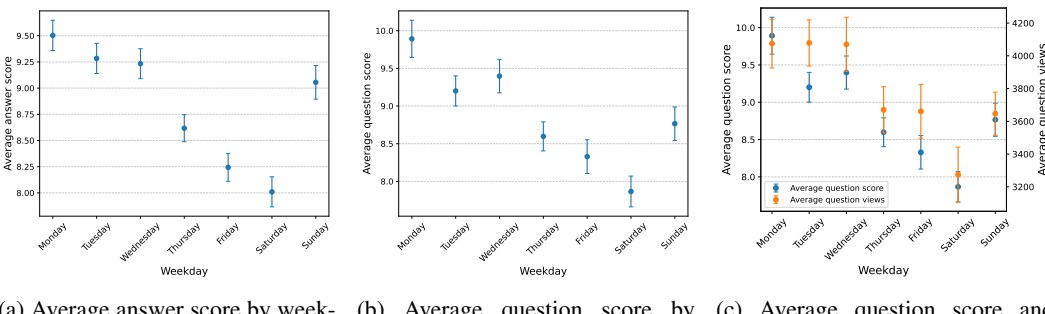

(a) Average answer score by weekday

(b) Average question score by weekday

(c) Average question score and views by weekday

Figure 1: Weekly temporal patterns in Stack Exchange engagement. User scores and views exhibit strong weekday effects, with higher engagement early in the week.

periods can substantially influence observed CTRs. Figure 2a illustrates these dynamics: average CTRs exhibit pronounced temporal variation across experimental training, observational training, and observational validation subsets. Monthly averages are highly correlated across subsets (pairwise correlations of 96–97%), indicating that these fluctuations are systematic rather than stochastic.

These findings raise concerns regarding the direct use of raw CTRs for model training. Temporal shifts in audience composition and preferences may dominate the signal, leading models to capture time-specific artifacts rather than structural attributes of headline quality. Figure 2b provides further evidence, showing substantial variation in impressions per package over time, including a marked increase in late 2023 coinciding with the U.S. election period. Such variation suggests non-stationarity in both traffic volume and audience characteristics. Consequently, models trained naively on observational CTRs risk conflating shifts in exposure and demand with causal effects of content, thereby limiting their ability to generalize beyond the training environment.

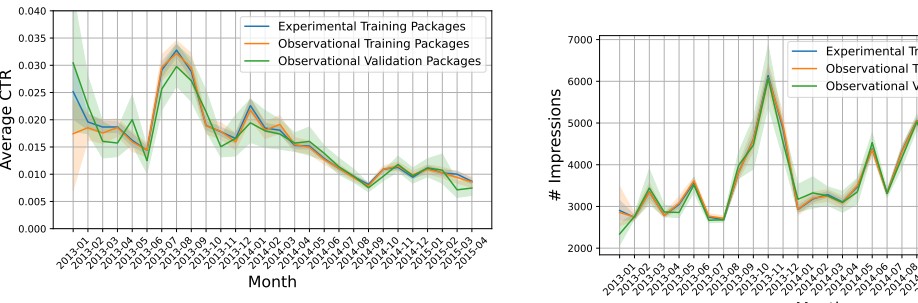

(a) Monthly average CTRs across different data splits.

(b) Monthly average impressions per package.

Figure 2: Temporal patterns in user engagement. The left plot shows variation in average click-through rates (CTR) across months, while the right plot shows the number of impressions per package, indicating changes in user traffic volume.

To compare learning from experimental vs. observational data, we fine-tune LLMs by adding a small head to the final embeddings: (i) a pairwise classification head with logistic loss to predict the higher-CTR headline (experimental), and (ii) a regression head with MSE to predict observed CTR (observational). In both settings, we use $L_2$ regularization and tune $\lambda$ on a validation set.

**Performance.** We evaluate all models on the same held-out set of headline pairs, where each pair comes from an A/B test with a known preferred headline. The evaluation objective is to assess whether the model correctly ranks the preferred headline higher. To do this, we compute the ROC AUC (Area Under the Receiver Operating Characteristic Curve), which reflects the model's ability to distinguish between better and worse-performing headlines. We use the Pythia suite of open-weight language models for all experiments Biderman et al. (2023). Figure 3a shows the ROC

AUC results for the Pythia-12B model. Training on the experimental dataset yields an AUC of 0.82, whereas the observational dataset produces a lower, but still above-chance, AUC of 0.74. This result is encouraging: it suggests that historical data, even without experimental variation and with only about 26% of the training packages, can still provide meaningful signals for preference learning. The performance gap also highlights the value of randomized feedback; exposure to counterfactual comparisons enables better generalization and more reliable preference estimation. Figure 3b further shows AUC improves with model size in both settings, yet the experimental–observational gap remains, underscoring the value of randomized feedback when available.

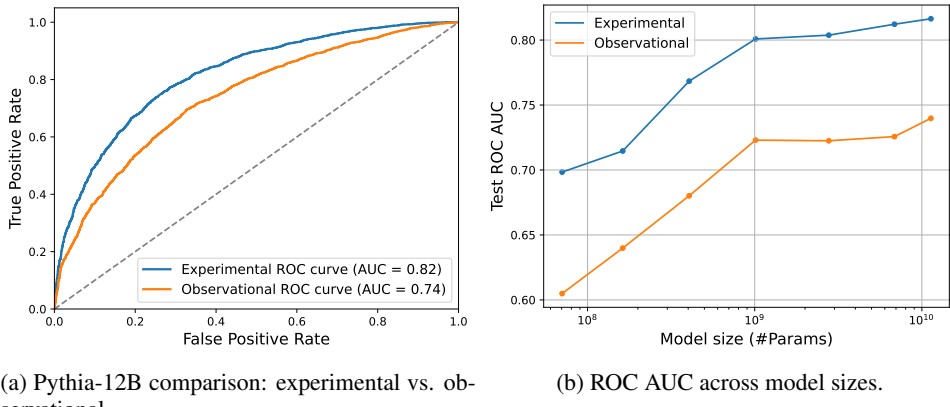

(a) Pythia-12B comparison: experimental vs. observational.

(b) ROC AUC across model sizes.

Figure 3: ROC on held-out Upworthy headline pairs. (a) ROC curves for models trained with experimental data outperform those trained on observational data. (b) Larger models yield better results in both settings, but the performance gap persists.

**Importance of regularization.** We study the role of regularization in observational learning and find that strong regularization is critical for generalization. As shown in Figures 4b and 4c, optimal validation loss occurs at $\lambda = 18,000$, yet the best test ROC AUC is achieved at $\lambda = 50,000$. This discrepancy suggests that in the presence of confounding factors, tuning hyperparameters solely based on validation loss may not suffice. The model may overfit to patterns influenced by spurious correlations in the validation data, rather than learning features that generalize causally to unseen headline comparisons. Figure 5a shows that this gap holds across model sizes: stronger regularization consistently yields better test performance than what validation loss would suggest. We further find that larger models generally require stronger regularization for optimal test performance. This observation implies that using a fixed regularization setting across models of different sizes is suboptimal. Figure 5b demonstrates this by plotting test performance against model size under fixed regularization levels. The figure shows a non-monotonic effect, larger models begin to overfit more if regularization is kept constant. These results emphasize the need to scale regularization appropriately with model capacity in order to maintain generalization, which is often overlooked in practice.

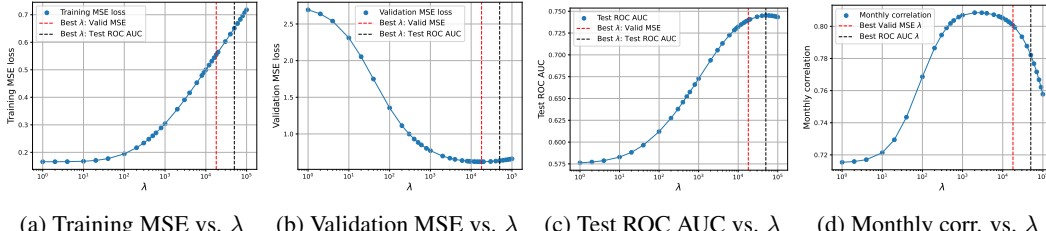

(a) Training MSE vs. $\lambda$    (b) Validation MSE vs. $\lambda$    (c) Test ROC AUC vs. $\lambda$    (d) Monthly corr. vs. $\lambda$

Figure 4: Effect of regularization strength ($\lambda$) on different evaluation metrics for the Pythia-12B model.

While these results underscore the critical role of regularization, they also raise a practical challenge when access to held-out experimental data for tuning hyperparameters is often limited. In

such cases, alternative strategies are needed to remove the effect of confounders. We address this issue in Section 4, where we introduce a method for explicitly correcting for confounding effects in observational fine-tuning.

**Temporal pattern overfitting.**    As discussed earlier, temporal variation in CTRs is a potential confounder in observational data. To assess how much models internalize these patterns, we compute the correlation between monthly average CTR estimates on the validation set and observed monthly CTRs in the training data. Figure 4d shows this correlation across values of $\lambda$ for the Pythia-12B model. We observe that moderate regularization improves alignment with temporal patterns, but higher regularization reduces it. Interestingly, the $\lambda$ that yields the best test performance comes well after this drop, indicating that suppressing temporal patterns helps the model on the causal evaluation of headlines. This trend holds across model sizes. As shown in Figure 5c, models consistently show lower temporal correlation at their optimal test-time $\lambda$, further suggesting that failing to effectively account for confounding patterns can impair generalization performance.

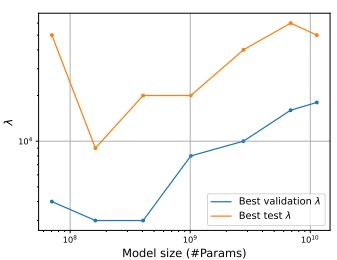 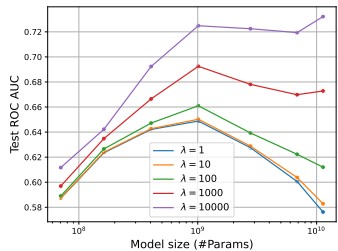 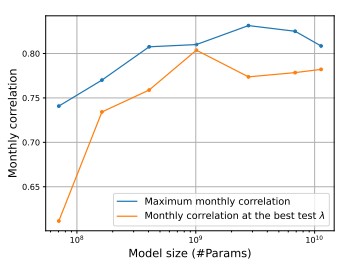

(a) Optimal $\lambda$ by model size (validation vs. test)
(b) Test ROC AUC with fixed regularization
(c) Monthly correlation

Figure 5: Regularization dynamics across model sizes. (a) Models require larger regularization to achieve optimal test performance. (b) Using a fixed regularization setting leads to non-monotonic scaling performance. (c) Models show lower correlation with temporal engagement patterns at their optimal test-time $\lambda$.

## 4    A CAUSAL FRAMEWORK FOR OBSERVATIONAL FINE-TUNING

We now introduce a formal framework to analyze the effect of confounding in observational fine-tuning. Suppose we have access to historical interaction data $\mathcal{D} = \{(\boldsymbol{X}_i, y_i)\}_{i \in \mathcal{I}}$, where $\boldsymbol{X}_i$ denotes the context vector for interaction $i$ and $y_i$ is the associated outcome (e.g., click-through rate or reservation rate). We assume that $\boldsymbol{X}_i$ can be decomposed into a textual decision variable $T_i$ (e.g., a generated title or headline) and auxiliary features $\tilde{\boldsymbol{X}}_i$, such that: $y_i = f(T_i, \tilde{\boldsymbol{X}}_i) + \epsilon_i$, where $f$ is the unknown outcome function and $\epsilon_i$ is zero-mean noise. The goal is to train a generative model $G : \tilde{\boldsymbol{X}} \to T$ that produces high-reward textual actions for new inputs. To capture confounding, we assume that the outcome function can be decomposed as:

$$y_i = g(T_i, \tilde{\boldsymbol{F}}_i) + h(\boldsymbol{C}_i) + \epsilon_i, \tag{1}$$

where $\boldsymbol{F}_i$ is the set of observed features including $T_i$, and $\boldsymbol{C}_i$ represents observed confounders that influence both the action and the outcome. The function $g$ captures the causal effect of the textual action and other features, while $h$ captures the contribution of confounders. Crucially, if $g$ and $h$ are entangled, estimating them independently may result in biased models and reward misspecification.

**Proposed method: DECONFOUNDLM**    Our proposed approach, Deconfounded Language Model Fine-Tuning (DECONFOUNDLM), involves first identifying and modeling the effect of confounders, and then explicitly removing their contribution from the observed outcomes. This allows the model to learn the causal impact of the textual input and other features, without being influenced by confounding effects. In our experiments, we apply an instrumental variable strategy to estimate the confounding component. However, the framework is flexible: other methods such as Double Machine Learning Chernozhukov et al. (2018) or Adversarial GMMs Dikkala et al. (2020) can be used,

provided the researcher is mindful of the assumptions of the methods. We further discuss the potential impacts and limitations in Appendix E.

**Example 1** (Partially Linear Regression). Consider a partially linear model where the confounder $p_i$, e.g., the price in Airbnb listings, enters linearly (Similar to the examples from Chernozhukov et al. (2018)):

$$y_i = g(T_i, \tilde{\boldsymbol{F}}_i) + \alpha p_i + \epsilon_i, \tag{2}$$

Here, $p_i \in \boldsymbol{C}_i$ is an observed confounder. In the Airbnb example, when optimizing titles to improve reservation rates, price may strongly affect $y_i$ and also correlate with certain title patterns (e.g., "affordable"). Estimating $g$ accurately thus requires adjusting for $p_i$ to avoid spurious correlations.

## 4.1 SIMULATION EXPERIMENTS

In this section, we turn to simulation experiments for a controlled evaluation of the proposed DE-CONFOUNDLM method. We base our simulation experiments on the MIND dataset (Wu et al., 2020), which contains over 160,000 English-language news articles with both titles and full text. We treat the article body as the input context and aim to generate a headline $T_i$ that maximizes a synthetic performance score $y_i$, interpreted as a proxy for engagement or click-through rate. To simulate realistic challenges in observational fine-tuning, we design two scenarios, *Orthogonal confounding* and *Entangled confounding*, where the observed outcome $y_i$ depends on both the textual quality and a confounding variable $p_i$ representing topic popularity (e.g., how much fan interest a team garners). We use the headline sentiment $s(T_i) = g(T_i, \tilde{\boldsymbol{F}}_i)$ as a measure of quality, as it is interpretable and easily measured.

Across both scenarios, we model the outcome as:

$$y_i = s(T_i) + 0.1 \, p_i + \nu_i, \tag{3}$$

where $p_i$ is the confounder, and $\nu_i \sim \mathcal{N}(0, 0.1)$ represents observational noise. We vary how $p_i$ is constructed across two settings:

- *Orthogonal confounding.* The confounder $p_i$ is independent of the sentiment $s(T_i)$, making its effect easier to isolate and remove. Specifically:

  $$p_i = \mathbb{1}(\text{title mentions West Coast team}) + 2 \cdot \mathbb{1}(\text{Central team}) + 3 \cdot \mathbb{1}(\text{East Coast team}) + \epsilon_i, \tag{4}$$

  where $\epsilon_i \sim \mathcal{N}(0, 0.5)$. This reflects a hypothetical bias where East Coast teams are generally more popular and draw higher engagement regardless of the title's quality.

- *Entangled confounding.* Here, popularity $p_i$ is correlated with the sentiment of the news abstract, mimicking a setting where emotional salience of the topic and engagement covary. For instance, sad events may draw more audience to the platform and lead to increased engagement. We model this with:

  $$p_i = \mathbb{1}(\text{title mentions West Coast team}) + 2 \cdot \mathbb{1}(\text{Central team}) + 3 \cdot \mathbb{1}(\text{East Coast team})$$
  $$- 10.5 \cdot s(\text{abstract}) + \epsilon_i. \tag{5}$$

**IV justification.** In this setting, both the abstract sentiment $s(\text{abs})$ and team mentions influence the latent popularity measure $p_i$, which in turn drives engagement $Y_i$. Omitting popularity from the analysis induces bias in the estimated effect of headline sentiment on $Y_i$, since part of the observed variation in engagement is mediated by shifts in audience composition or topical salience. Instrumental variables (IV) address this problem by exploiting variation that affects the outcome only through its impact on popularity. In our case, team mentions satisfy this requirement: they generate exogenous shocks to popularity (e.g., attention surges around specific sporting events) but do not otherwise alter the causal path from headline sentiment to engagement.

The IV procedure estimates and removes the contribution of popularity, leaving only the component of engagement that is causally attributable to headline sentiment. This illustrates the classical

conditions under which IV estimation is effective: (i) *relevance*, since team mentions are strongly correlated with popularity, and (ii) *exclusion*, since they affect engagement only through popularity. When these conditions are satisfied, IV recovers the true causal effect of headline sentiment even in the presence of confounding.

**Comparative methods.** We evaluate seven approaches: (1) a base pre-trained model, (2) supervised fine-tuning (SFT), (3) RL with access to ground-truth sentiment (which serves as a baseline) (4) RL using observed performance without controlling for confounders, (5) RL models that incorporate popularity either as input text or as a scalar feature in the final layer, (6) ODIN (Chen et al., 2024) as an example of a method that relies on counterfactual invariance, and (7) our proposed method DECONFOUNDLM-IV, which estimates and removes the confounder effect using an instrumental variable.

**Results.** We evaluate all models' generations after the RL step on a held-out set of 3,000 news articles. Table 1 summarizes the average sentiment of generated headlines and the frequency of team name mentions, which serve as a proxy for reliance on the popularity-based confounder. In the *Orthogonal* setting, the model trained on observed performance (without accounting for confounding) is able to improve headline sentiment, indicating that it learns part of the true signal. However, it also shows a marked increase in the frequency of team name mentions, suggesting reliance on popularity cues. Incorporating popularity information, either via text prompts or as an input feature, reduces this effect. Among all methods, DECONFOUNDLM-IV more closely matches the sentiment gains of the true-reward model while not generating unnecessary references to team names caused by the confounding variable.

Table 1: Comparison of models under two confounding scenarios. The table reports the mean sentiments and the number of generated titles mentioning teams by region. Models are tested on 3,000 headline generations. Note that the reported results for the first four models are identical across both scenarios, as they do not rely on the observed performances; the difference between the two scenarios lies solely in how the observed performance is constructed.

| Model | Scenario 1: Orthogonal | | | | Scenario 2: Entangled | | | |
| --- | --- | --- | --- | --- | --- | --- | --- | --- |
| | Sent. | W | C | E | Sent. | W | C | E |
| Base Pre-Trained Model | 0.684 | 177 | 795 | 717 | 0.450 | 177 | 795 | 717 |
| SFT Model | 0.716 | 159 | 634 | 610 | 0.716 | 159 | 634 | 610 |
| Model with only sentiment | 0.950 | 165 | 714 | 728 | 0.960 | 187 | 697 | 720 |
| Model with sentiment + noise | 0.934 | 172 | 699 | 664 | 0.958 | 174 | 716 | 699 |
| RL w/ observed performance | 0.956 | 214 | 914 | 1016 | 0.735 | 186 | 908 | 1041 |
| RL w/ pop. in text | 0.932 | 158 | 655 | 638 | 0.718 | 166 | 676 | 629 |
| RL w/ pop. in layer | 0.935 | 178 | 679 | 661 | 0.800 | 189 | 829 | 835 |
| ODIN (Chen et al., 2024) | 0.934 | 178 | 729 | 680 | 0.807 | 249 | 1018 | 1063 |
| **DECONFOUNDLM-IV** | 0.939 | 181 | 692 | 682 | 0.937 | 170 | 736 | 694 |

The *Entangled* case presents a more challenging scenario. Here, the naive model trained on observed performance fails to improve sentiment and heavily generates team names. While models that include popularity in the input text or final layer performed well in the *orthogonal* setting, they struggle to recover the sentiment-performance relationship in this setting. ODIN also achieves a similar performance in terms of sentiments of the generated headlines, but generates many more team names. In contrast, DECONFOUNDLM-IV demonstrates strong robustness. It successfully suppresses the influence of the confounder and generates headlines with **16%** higher sentiment scores compared to the next highest sentiment scores. Full experimental details for the simulation experiments are provided in Appendix D.

**Reward-sentiment correlation.** We now turn to the question of why DECONFOUNDLM delivers stronger results: How closely do the reward models actually follow the true sentiment measures? Table 2 shows the average Pearson correlation between predicted rewards and sentiment scores in the reward validation set, under two confounding scenarios. In the orthogonal case, observed

performance is positively correlated with sentiment, allowing most models to achieve a positive correlation between their reward estimates and sentiment. However, in the entangled case, where the confounder (e.g., team popularity) effect is entangled with the outcome, this relationship breaks down. Most of the models that do not account for the confounder, or attempt to include it through text features or final-layer embeddings, fail to maintain a positive correlation between predicted rewards and sentiment. In contrast, DECONFOUNDLM-IV remains robust across both scenarios, maintaining a strong positive correlation.

Table 2: Correlation between predicted rewards and sentiment across two confounding scenarios. Each cell shows the Pearson correlation on the train and validation sets. The reported results for the first two models are identical across both scenarios, as they do not rely on the observed performances.

| Model | Scenario 1: Orthogonal | | Scenario 2: Entangled | |
|---|---|---|---|---|
| | Train | Valid | Train | Valid |
| Model with only sentiment | 0.913 | 0.872 | 0.913 | 0.872 |
| Model with sentiment + noise | 0.910 | 0.891 | 0.807 | 0.802 |
| RL w/ observed performance | 0.880 | 0.859 | -0.073 | -0.079 |
| RL w/ pop. in text | 0.839 | 0.827 | -0.261 | -0.253 |
| RL w/ pop. in layer | 0.862 | 0.841 | 0.623 | 0.627 |
| ODIN Chen et al. (2024) | 0.654 | 0.641 | -0.536 | -0.542 |
| **DECONFOUNDLM-IV** | **0.915** | **0.886** | **0.898** | **0.867** |

## 5 CONCLUSION AND DISCUSSION

Our findings suggest that using historical data to fine-tune language models can be a double-edged sword: while it provides valuable information without the need for experimentation, it could also introduce the risk of learning from confounded outcomes. Through both real-world and synthetic experiments, we show that models trained on observational data may internalize spurious correlations that are not causally linked to content quality. To mitigate this, we introduce DECONFOUNDLM, a method that explicitly adjusts for observed confounders in the fine-tuning process. By separating confounding influences from the outcome signal, our approach enables more causally grounded learning without relying on the counterfactual invariance assumption that is often used in prior work. Across multiple settings, we find that DECONFOUNDLM improves fine-tuning outcomes and better captures the true effects of textual inputs. Finally, while our primary focus is performance and causal inference, we note that confounding can also introduce fairness concerns. If unaddressed, it may lead models to replicate or amplify structural biases in the data. We view causal deconfounding as a promising direction for aligning language models not only with user preferences but also with broader values of equity and accountability.

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

## A   PROBLEM SETUP AND BACKGROUND

Fine-tuning LLMs to align their outputs with user preferences is a common approach for enhancing their performance. This process typically relies on *labeled preference data*, which may be collected through human annotations Ziegler et al. (2019), automated feedback mechanisms (e.g., RLAIF Lee et al. (2024)), or structured reasoning tasks (e.g., Guo et al. (2025)). Two major paradigms are commonly used to incorporate preference data into LLM training: (i) Reward Modeling followed by Reinforcement Learning, and (ii) Direct Preference Optimization.

In the former, a reward model $r_\phi(x, y)$ is first trained to predict human preferences between outputs given an input $x$. The model is typically trained using pairwise comparisons, optimizing a Bradley-Terry likelihood:

$$\mathcal{L}_{\text{RM}}(\phi) = -\mathbb{E}_{(x, y_w, y_l) \sim \mathcal{D}} \left[ \log \sigma \left( r_\phi(x, y_w) - r_\phi(x, y_l) \right) \right],$$

where $\sigma$ is the logistic sigmoid. Once trained, this reward model is used to fine-tune the language model $\pi_\theta(y \mid x)$ using reinforcement learning algorithms such as Proximal Policy Optimization (PPO), which maximize expected reward while regularizing against a reference policy:

$$\max_{\pi_\theta} \mathbb{E}_{x \sim \mathcal{D}, \, y \sim \pi_\theta(\cdot | x)} \left[ r_\phi(x, y) - \beta \, \text{KL} \left( \pi_\theta(\cdot \mid x) \, \| \, \pi_{\text{ref}}(\cdot \mid x) \right) \right].$$

In contrast, *Direct Preference Optimization (DPO)* bypasses reward modeling entirely and directly updates the policy to prefer higher-rated responses using a contrastive objective over preference pairs Rafailov et al. (2024b).

In industrial applications such as optimizing click-through rates for headlines, boosting booking rates on rental platforms, or improving adherence in health messaging, the gold standard for evaluating outcomes is randomized controlled trials, which allow unbiased estimation of causal effects. However, such experiments are expensive and often infeasible in practice. Meanwhile, organizations often have abundant *observational data*: historical logs of content such as page titles, messages, or headlines and their associated outcomes. This data can be used directly for fine-tuning, either by training a reward model or by constructing preference pairs for methods like DPO. The challenge, however, is that observational data is subject to confounding: unobserved variables may influence both the textual content and the observed outcome, leading to spurious correlations. For example, consider an LLM deployed at Airbnb to generate listing titles aimed at increasing reservation rates. Historical data may show that listings with the word *"affordable"* in the title perform better. However, this could reflect the underlying confounding effect of the price, as lower-priced listings generally get higher reservations. A model fine-tuned naively on this data may learn to associate *"affordable"* with success in all contexts, leading to unsuitable generations. Say you are asking the model to generate a title for a luxury riverside property, and the model generates ``Affordable log chalet { perfect for solo travelers''!

This is a clear instance of reward misspecification: the model learns to optimize a proxy signal that only partially reflects the true objective. While prior work Gao et al. (2023); Rafailov et al. (2024a)

has investigated reward over-optimization in both classical RLHF and Direct Alignment settings, these studies have largely focused on empirical scaling behavior and optimization dynamics. In this work, we take a step further by analyzing the role of confounding and causal misalignment in fine-tuning large language models using observational data. In the following section, we investigate how relying on historical preference data can lead to biased and potentially flawed model fine-tuning, by examining a real-world case study.

## B THE MONDAY EXPERIMENT: AN EXAMPLE OF CONFOUNDING IN OBSERVATIONAL DATA

In this section, we present an example illustrating how confounding can lead to reward misspecification when fine-tuning language models using historical data. Specifically, we construct a dataset similar to that used by Askell et al. (2021), based on data from the Academia Stack Exchange. In their study, the authors fine-tune a question-answering model and, in one step of their fine-tuning, Preference Model Pre-Training (PMP), use historical data to guide learning. They treat answer scores as preference signals and train the model to prefer higher-scored answers in cases where multiple answers are available for a question. This PMP step is followed by fine-tuning with human feedback, to ensure the alignment of the model's preferences with human judgments. In our experiment, we investigate what happens when human feedback is unavailable and only observational data is used for fine-tuning.

While using the scores can signal which answer is more helpful, these scores are not the outcomes of randomized experiments, rather could be affected by user engagement patterns. For example, answers posted earlier may receive more views and thus more votes. One confounder we investigate is periodicity in platform engagement across different weekdays. To investigate this, we analyze the average answer scores by weekday. As shown in Figure 1a, answers posted on Mondays receive significantly higher scores than those posted on Fridays. While one might speculate that this might be causal and reflect differences in writing quality, we observe a similar pattern in the average scores of questions themselves (Figure 1b), suggesting that broader engagement trends may be at play.

We further examine the number of views per question as a proxy for user exposure. Since the dataset does not include view counts for individual answers, we cannot directly assess the effect of exposure at the answer level. Figure 1c displays both question views and scores over time. The strong correlation between the two suggests that the observed temporal trends are more likely driven by fluctuations in user activity than by differences in content quality.

To test whether this bias can influence model behavior, we simulate a fine-tuning setup similar to that of Askell et al. (2021). We construct answer pairs based on user scores and designate the higher-scoring answer as preferred. For answers posted on Mondays, we prepend a neutral ``Happy Monday!'' phrase to introduce a content marker correlated with engagement rather than quality. We then evaluate model generations on 3000 held-out questions and count how often the words "Happy" and "Monday" appear. Table 3 summarizes the results. The base pre-trained model rarely generates these terms. The supervised fine-tuned (SFT) should ideally capture the distribution in the data (generation temperature is set to 1). Our results show a frequency of 13.7% ± 0.2% for the first word "Happy" which is consistent with the distribution in the data ($\sim 1/7$). In comparison, the DPO model generates "Happy" in 21.2% ± 0.8% and "Monday" in 11.8% ± 0.8%, representing substantial increases of approximately 7.5 and 2.7 percentage points, respectively. To assess whether these increases are statistically significant, we perform independent two-sample t-tests over the 25 generation rates from each model. The difference in "Happy" usage is highly significant ($p = 6.5 \times 10^{-10}$), and the increase in "Monday" usage is also statistically significant ($p = 3.4 \times 10^{-3}$). These results show that the model has internalized and amplified a spurious temporal signal. Additional details of this experiment, as well as further details about data and training characteristics, are provided in Appendix B.

This case highlights how confounding variables in observational datasets can lead to reward misspecification and unintended behavior in fine-tuned models. Without accounting for causal structure, models may learn to exploit spurious signals that correlate with success, even when they do not contribute to genuine task quality.

Table 3: Mean percentage (standard error) of generations containing "Happy" and "Monday" across 5 generation seeds for the base model, and 25 runs (5 fine-tuning seeds × 5 generation seeds) of SFT and DPO fine-tuning. DPO fine-tuning significantly amplifies the spurious weekday signal.

| Model | Generations per Run | Num. Runs | With "Happy" (%) | With "Monday" (%) |
|---|---|---|---|---|
| Base Model | 3000 | 5 | 1.13 (0.04) | 0.07 (0.02) |
| SFT Model | 3000 | 25 | 13.69 (0.21) | 9.05 (0.19) |
| DPO Model | 3000 | 25 | **21.22 (0.77)** | **11.78 (0.81)** |

**Data.** To replicate and extend the setup of Askell et al. (2021) we use data from the Academia Stack Exchange. The dataset contains 104,426 question-answer pairs. We retain only those questions with multiple answers, reducing the data to 82,737 answer instances. To reduce memory usage during training, we further restrict to questions and answers with fewer than 180 words, yielding 33,194 answers across 14,319 questions. Of these, 4,937 answers were written on a Monday.

We split the questions into three groups: 5,000 for supervised fine-tuning (SFT), 3,000 for testing, and the remainder for reward-based fine-tuning. For each question in the fine-tuning subset, we form ordered answer pairs by comparing scores and labeling the higher-scored answer as preferred. We cap the number of pairs per question at 10 to prevent imbalance. This yields 11,886 pairs, where we find a notable weekday skew: in pairs with only one Monday answer, 958 have the Monday answer as preferred, while 887 have it as rejected, hinting at temporal confounding.

**Fine-tuning setup.** We use the 360M parameter `SmolLM2-Instruct` Allal et al. (2025) model as the base and perform two-stage fine-tuning.

**Supervised Fine-Tuning (SFT).** The SFT step uses the answer text as the assistant response and the corresponding question as input. Training is done for 1 epoch with a batch size of 8 and a learning rate of $2 \times 10^{-4}$ using the AdamW optimizer (8-bit). We apply LoRA Hu et al. (2022) with rank 16 and dropout 0.1. Inputs are tokenized using a custom prompt template with a 512-token sequence limit.

**Direct Preference Optimization (DPO).** The DPO stage initializes from the SFT checkpoint and fine-tunes using the constructed answer preference pairs. We use a $\beta$ of 0.1 and train for up to 4 epochs with a batch size of 8. LoRA is applied with rank 8. The maximum prompt and completion lengths are 256 and 512 tokens, respectively.

Generations for evaluation are performed on a held-out set of 3,000 questions, and model outputs are assessed for lexical artifacts. For the base model, which is fixed and not subject to any fine-tuning variability, we introduce randomness only through the generation process by using 5 different random seeds. In contrast, both the SFT and DPO models are subject to randomness in fine-tuning as well as generation. Specifically, we fine-tune each model using 5 different random seeds for initializing the model head, and then generate outputs from each trained model using 5 different generation seeds. This results in 25 runs per model, each producing 3000 generations.

To qualitatively assess the effect of fine-tuning, Table 4 presents representative generations from the base, SFT, and DPO models for a single held-out question. This example illustrates how fine-tuning shapes model behavior and provides evidence that the fine-tuning process is likely implemented correctly. The base model produces a generic answer formatted in Markdown (e.g., using `**` for bold), reflecting conventions likely present in its pretraining data. In contrast, the SFT model aligns more closely with the domain-specific formatting patterns found in the training set, namely, HTML tags. It adopts a tone more typical of answers on the target forum. The DPO model builds upon this, showing further alignment with patterns found in high-scoring answers: it employs richer formatting (e.g., ``) and incorporates external references, likely reflecting learned associations between such features and answer quality.

**Compute.** All experiments for this setup were conducted on an NVIDIA RTX A6000 GPU. The Supervised Fine-Tuning (SFT) stage is relatively lightweight and completes in approximately 10 minutes. In contrast, the Direct Preference Optimization (DPO) stage is more computationally intensive due to its iterative training on preference pairs and takes around 1 hour to run per seed.

Table 4: Comparison of model responses for a given question.

| Question | Model | Generated Answer |
|---|---|---|
| "<p>I've been a Computer Sciences student for 4 years now, give or take, and my experience with emailing teachers/professors (teachers from now on) has been either of two cases: </p> <ul> <li>they reply to my entire email</li><li>they read the last sentence or paragraph and reply to that.</li></ul> <p>Generally, if a teacher fits in one of those categories, they always respond like that.
However, for me it's quite frustrating to receive an email back (within 5 minutes, an hour, a day) with a reply, to only one point, and then having to mail back with my other questions, removing one question at a time.</p> <p>To note: my e-mails are always very respectful, neatly written, and without spelling or other mistakes.</p> <p>What can I do to get a higher percentage of questions-answered-per-email response?</p>" | **Base Model** | 1. **Include specifics about the difficulties you face**: Instead of a general statement, explain the difficulties you're facing. This will show that you're willing to help by providing more details. 2. **Offer solutions to the problem**: Provide some strategies or tips to help teachers improve their email responses to your questions. This will show that you're proactive and interested in finding a solution to the problem. 3. **Provide clear examples of your goals and objectives**: Be specific about what you're trying to achieve by improving the response rate. For instance, you could say that you'd like to see more direct responses or more personalized responses from the teachers. 4. **Highlight any potential benefits to the school**: While you mentioned that the teacher-student ratio might be a problem, you could also mention that the improvement in response rate could positively affect the students, such as by having more individualized support. 5. **Emphasize the time and effort required**: Acknowledge that sending multiple emails back and forth might be time-consuming, but also express that the benefits of getting more responses are worth the extra effort. Example of how you could write a specific paragraph to demonstrate this: "I'm frustrated that I've received responses to my |
| | **SFT Model** | <p>Happy Monday!</p><p>If my general attitude is that you should be able to reply to a very large number of emails with a single email, your teacher should answer the first few emails, since they don't need to spend a lot of time answering those.</p><p>On the other hand, if your teacher is very particular about your answers, you might want to offer more details about what you're trying to accomplish in your course, and what kind of questions you have and are trying to get answered.</p> |
| | **RL (DPO)** | <p>Happy Monday!</p><p>The first point is that people generally answer more questions than they reply to in the course of an email. In a rough order, the answers are:</p><ul><li>Less frequently, but they read the last sentence or paragraph and then reply to that: <a href="http://en.wikipedia.org/wiki/Reciprocity"> Reciprocity</a> with you. It's nice to hear from you again and it's worth the extra effort for them to do so.</li><li>Less frequently, but you read the last sentence or paragraph and then reply to that: <a href="http://en.wikipedia.org/wiki/Reciprocity"> Reciprocity</a>.</li><li><a href="https://stackoverflow.com/ask" target="_blank">Ask a question</a> and get answers.</li><li>Less frequently, but you don't read the last sentence or paragraph and reply to that: ... |

## C  UPWORTHY EXPERIMENT DETAILS

We follow a similar data processing approach to that of Ye et al. (2024), using the Upworthy dataset. The full dataset includes 150,817 headline-image "packages" across 32,487 A/B tests. Since some tests involve variation in both headlines and images, we restrict our analysis to headline-only tests where the image remains fixed. This filtering yields 17,682 headline-only tests comprising 77,245 packages.

To construct the experimental dataset, we generate all possible headline pairs within each A/B test and retain only those with a statistically significant difference in click-through rate (CTR) at the 5% level. This results in 41,624 headline pairs covering 27,745 packages. We split these into training (60%), validation (20%), and test (20%) sets, while ensuring no headline appears in more than one split to avoid data leakage. The final dataset includes 24,842 training pairs, 8,395 validation pairs, and 8,387 test pairs.

These statistically significant pairs form the basis of our experimental setting. To simulate a non-experimental setting, we derive a corresponding observational dataset. For each headline test in the training set, we randomly retain only one package and discard the counterfactual. This results in 8,499 training packages, representing approximately 26% of the total packages. This setup reflects a typical historical logging scenario, where only observed outcomes are available. Table 5 provides summary statistics of the experimental and observational datasets.

Before moving on to the modeling details, we briefly highlight a potential confounder that can affect observational CTRs: temporal variation in user engagement and topic salience. The probability that a user clicks on a given package depends not only on the quality or attractiveness of the headline, but also on who the viewers are and how relevant or important the topic is at the time. For instance, if a major sporting event occurs, the site may receive a surge of sports fans, whose preferences disproportionately influence overall CTRs. Similarly, politically themed headlines may receive more engagement during election periods. Figure 2a shows the average CTR by month for three data subsets: experimental training packages, observational training packages, and observational validation packages. We observe a clear temporal pattern, with certain months getting substantially higher CTRs. Moreover, the CTR trends are highly correlated across subsets (pairwise correlations of monthly averages are between 96% and 97%), suggesting that these fluctuations are systematic rather than random.

This raises a concern for models trained directly on raw CTRs. They may overfit to superficial, time-related artifacts rather than learning meaningful properties of headline quality. As previously discussed, variations in CTR may partly reflect changes in the user population and taste rather than differences in content effectiveness. Figure 2b provides evidence of these changes, showing substantial variation in the number of impressions per package across months. Notably, there is a marked increase in impressions toward the end of 2023, coinciding with the U.S. election period. These fluctuations suggest that the volume and potentially the composition of website traffic change over time. As a result, shifts in user demographics or interests could introduce biases into the observed CTRs, potentially misleading models trained on such observational data.

**Reward modeling.**  To train reward models, we use a prompting structure where the model is asked to generate a headline for a given news abstract:

```
System: You are an editor of a news website.
Your task is to generate a headline for each news article that
will attract the most readers. The headline should be less than 40 words.
Only respond with the headline.
User: The news abstract is '{lede}' News posted at {created_at}
Assistant: {headline}
```

We use models from the Pythia suite Biderman et al. (2023) to generate embeddings, specifically extracting the representation of the final token in each output. A classification or regression head is added on top of this embedding to predict outcomes (CTR or preference), and an $L_2$ regularization parameter $\lambda$ is tuned to manage overfitting, as detailed in the main text.

**Compute.**  The most computationally intensive part of this experiment is generating embeddings using models from the Pythia suite. We extract the final-token representations, which serve as in-

Table 5: Summary statistics of the Upworthy dataset for experimental and observational settings.

| Statistic | Upworthy Data | |
|---|---|---|
| Total headline-only A/B tests | 17,682 | |
| Total packages | 77,245 | |
| | **Experimental Data** | **Observational Data** |
| Statistically significant pairs | 41,624 | – |
| Packages in significant pairs | 46,330 | – |
| Training pairs | 24,842 | – |
| Training packages | 27,745 | 7,285 |
| Validation pairs | 8,395 | – |
| Validation packages | 7,527 | 2,079 |
| Test pairs | 8,387 | 8,387 |

puts to the reward models. These embedding computations are performed on an AMD Radeon 7900 GPU. For the largest model used in our experiments, Pythia-12B, the embedding generation takes approximately 12 minutes for the observational dataset and about 1.5 hours for the experimental dataset, which is larger. Once embeddings are obtained, training the reward models with a classification or regression head is relatively lightweight and runs efficiently on the Intel(R) Xeon(R) Gold CPU @ 2.90GH.

## D    DETAILS OF SIMULATION EXPERIMENTS

For our synthetic experiments, we use the MIND (Microsoft News Dataset) (Wu et al., 2020), which contains 160,000 English news articles, each with a headline and article body. To simulate user engagement, we construct synthetic performance scores (interpretable as click-through rates) for the article headlines using equations equation 3, equation 4, and equation 5. To find the sentiment of each headline in the data, we use the sentiment analysis model from Yuan (2023).

To ensure domain consistency, we focus on the sports category, which includes 54,553 articles, the largest among all categories. The data is split as follows: 20,000 articles for Supervised Fine-Tuning (SFT), 10,000 for Reward Modeling (RM), 3,000 for reward validation, 10,000 for Proximal Policy Optimization (PPO), and the rest for testing.

These synthetic scenarios allow us to explicitly test whether models can recover the true effect of sentiment when the observed performance signal is partially corrupted by a structured confounder.

**Supervised Fine-Tuning.**    We fine-tune a language model using SFT, where the model is prompted to generate engaging headlines from article abstracts. The prompting structure is:

```
System: You are an editor of a news website. Your task is to
generate a headline for each news article that will attract the most
readers. The headline should be less than 30 words. Only respond with
the headline.
User: The news abstract is '{abstract}'
Assistant: {headline}
```

The base model is `HuggingFaceTB/SmolLM2-360M-Instruct`, fine-tuned with LoRA (rank 16, $\alpha$=32, dropout=0.1) for one epoch. We use a learning rate of $2e-4$ and batch size of 8.

**Reward modeling.**    To train reward models on the synthetic performance scores, we perform hyperparameter tuning over several learning rates: {2e-4, 6e-4, 8e-4, 1e-3, 2e-3}. Based on prior findings (Ouyang et al., 2022), we limit training to one epoch to avoid overfitting.

**Generation results.** Table 1 summarizes the average sentiment of generated headlines and the frequency of team name mentions, which serve as a proxy for reliance on the popularity-based confounder. We discussed these results in Section 4.1.

**Compute.** Our simulation experiments were run using two types of GPUs: NVIDIA RTX A6000 and AMD Radeon 7900. For each combination of training seed and learning rate, reward modeling takes approximately 3–5 minutes on either GPU. However, the PPO fine-tuning stage is significantly more time-consuming, requiring about 2–3 hours to complete per setting.

## E  IMPACTS AND ASSUMPTIONS OF OUR FRAMEWORK

Our framework enables the use of observational data to align large language models (LLMs), thereby opening new possibilities for alignment with significant potential for positive social impact. As discussed in the main body of the paper, there are many real-world scenarios where conducting randomized experiments on content and messaging is infeasible, while firms often possess extensive historical observational data. In such cases, leveraging this data can substantially improve the alignment of LLMs with organizational or societal objectives. Consider, for example, a messaging system designed to improve medication adherence among patients. While running an experiment might be challenging due to engineering and ethical challenges, optimizing such a system using observational data could lead to substantial improvements in health outcomes. However, as with any machine learning paradigm that seeks to optimize a performance metric, this approach also presents challenges. As highlighted in prior work Mehrabi et al. (2021), various forms of bias can influence the outputs of machine learning models.

Our framework specifically targets biases arising from confounders that influence both the treatment and the outcome. While we have not yet conducted empirical evaluations of the bias correction component with respect to mitigating group-level disparities, the proposed method can be used to account for societal factors that might otherwise lead a model to prefer one textual input over another based on irrelevant or unfair criteria. Furthermore, researchers and practitioners must consider heterogeneity in individual responses to different texts to prevent the model from unintentionally encoding or amplifying structural disparities. For example, in a mobile health messaging application, if a particular message yields high adherence overall but performs poorly for a specific subgroup, it is crucial to incorporate recipient characteristics into the model to ensure equitable outcomes and avoid disproportionately favoring majority groups.

Turning to the theoretical underpinnings of our framework, prior work (see Section 2.3) often assumes that confounders have no effect on the outcome, implying a functional form $f(\boldsymbol{F}, \boldsymbol{C}) = f(\boldsymbol{F})$. However, this assumption may not hold in practice, especially in business settings where variables such as price are important drivers of outcomes. In contrast, our approach allows for a more realistic representation of the data-generating process, formulated as follows:

$$
\begin{aligned}
y_i &= f(\boldsymbol{X}_i) + \epsilon_i \\
&= f(T_i, \tilde{\boldsymbol{X}}_i) + \epsilon_i \\
&= g(T_i, \tilde{\boldsymbol{F}}_i) + h(\boldsymbol{C}_i) + \epsilon_i.
\end{aligned}
\tag{6}
$$

This formulation allows confounders to have a meaningful effect on outcomes, rather than assuming that outcomes are independent of confounder values. To ensure tractability, we impose two assumptions within our framework. First, we assume exogeneity of the error term conditional on the observed covariates, that is, $\mathbb{E}[\epsilon_i | \boldsymbol{X}_i] = 0$. This assumption is commonly made in empirical research involving high-dimensional covariates Zou & Zhang (2009), though it is not without limitations. As discussed in Fan & Liao (2014), even in high-dimensional settings, incidental or unintentional endogeneity can arise due to selection bias or model misspecification. Second, we assume a separable functional form in the final line of Equation 6, in which the effects of the confounders and the remaining variables are additively decomposed. Importantly, the model remains flexible enough to capture interactions between $g$ and $h$ through their shared inputs, as illustrated in the entangled case described in Section 4.1.

While our framework introduces greater flexibility than prior approaches, we acknowledge the limitations of these assumptions. The authors are currently working on developing a more general framework that further relaxes these conditions.

