# OpenReview forum: "Learning from Observational Outcomes: Toward Causally-Aligned Language Model Fine-Tuning"
_ICLR.cc/2026/Conference — ICLR 2026 Conference Withdrawn Submission_

### Official Review · Reviewer_tWw4 · 2025-10-27

**Soundness:** 2
**Presentation:** 1
**Contribution:** 2
**Rating:** 2
**Confidence:** 4

**Summary:**

The paper addresses the challenge of using large language models (LLMs) to optimise a specific task objective (reward) when only historical observational data are available for post-training. In such settings, distribution shifts between the observational data and the deployment environment—often caused by confounding variables (e.g. seasonality)—can bias the learned reward signal. As a result, directly fine-tuning the LLM on this observational data risks reinforcing spurious correlations rather than genuine causal relationships. To mitigate this, the authors propose a method to estimate a “deconfounded” reward function using an instrumental variable (IV) approach. This procedure aims to remove the influence of confounding factors and recover the causal effect of model actions on the true reward. The LLM is then (I believe) trained using this deconfounded reward estimate, rather than the reward observed in the historical dataset. Through empirical evaluation on a semi-synthetic dataset, the authors demonstrate that this IV-based correction leads to improved downstream performance and more robust generalisation to the true deployment environment.

**Strengths:**

- Using observational data for the purpose of LLM training and fine-tuning is an important problem with many real world applications.
- The authors provide many datasets and examples where training on observational data could lead to performance benefits.

**Weaknesses:**

1. **Lack of details:** My biggest issue with this paper is that the proposed method and experiments (particularly in Section 4) are not described in details sufficient to fully understand the contributions of this work, evaluate their validity and ensure the reproducibility of results. Most importantly:
    - What causal assumptions are imposed on the set of confounders for DefoncoundLM to work? (See also the question below)
    - What does DeconfoundLM really entail? How exactly is the contribution rom the observed outcomes “removed” (l. 320)?
    - After it has been removed, why happens next? Is the “confounder-adjusted reward” used as a reward function for RL?
    - If RL is used for post-training, what kind of optimisation algorithm is used? PPO? GRPO?
    - Does the method involve training a linear probe on the LLM embeddings? If yes, what is the architecture of this probe? If no, how is the reward predicted in Table 2?
    - For the baselines described in l. 383, how are the baselines exactly constructed and what loss was used for training?
I also provide additional questions in the ‘Question’ section below. I consider providing exhaustive answers to these questions a necessary step towards improving the paper, without which it is very difficult for me to evaluate the quality of this work.
2. **Limited novelty:** Results in Section 3 seem to validate the standard principles of the causal modelling: if you do not account for confounding in your modelling framework, your results will learn spurious correlations that will not generalise to the randomised (rather than observational) setting. The results in this paper seem to validate that this is also the case when fine-tuning language models (or training regression/classification heads on their embeddings), showing that LLMs follow the training dynamic imposed by the non-causal loss function, as expected. While this validation is nice, it provides limited novel insight or contributions.
3. **Limited evaluations:** The proposed method is only evaluated on a single semi-synthetic dataset. More dataset, with different relationships between confounders and the outcome function, would be necessary to further see the level of improvement the method can provide, and further illuminate under which circumstances it provides most benefits.

**Questions:**

- In line 255 you say claim that ‘the results emphasise the need to scale regularisation appropriately with model capacity to maintain generalisation’. However, in your results in Figure 5b, isn’t it the case that $\lambda=10000$ provides best performance across all model sizes? If yes, where is the above conclusion coming from?
- In the paragraph opening section 4, I think there is some confusion around the notation. In particular, what is the relationship between the auxiliary features $\tilde{\mathbf{X}}_i$, the observed features $\tilde{\mathbf{F}}_i$ and the confounders $\mathbf{C}_i$? Are they overlapping or disjoint sets? A causal diagram would be very helpful in understanding the relationships between different variables. Further, to make this estimation problem well-defined, it would be great to state what exact assumptions are imposed on the set of confounders and the general observational data. For example, is the set of observed confounders sufficient to disentangle the effect of the textual “action” on the outcome Y (unconfoundedness)? Answering this question rigorously and in detail would also make it clearer under which circumstances we can expect most gains from using this method.

---

> ### Author Response · Authors · 2025-12-03
>
> We thank the reviewer for carefully reading our paper and for emphasizing both the importance of using observational data for LLM fine-tuning and the practical relevance of the applications we discuss. We are glad that the motivation resonated with you and that you found the examples and datasets illustrative of the potential value of observational signals.
>
> Below, we address your concerns point by point and will incorporate the corresponding clarifications and details in the revised manuscript.
>
> 1. **Lack of details about DeconfoundLM and the experimental setup**
>
> (a) Causal assumptions and confounders
>
> The paper currently introduces the structural outcome model in Section 4 and Appendix E, where we write $y_i = g(T_i, \tilde F_i) + h(C_i) + \epsilon_i,$
> with $T_i$ representing the textual action, $\tilde F_i$ the auxiliary (non-confounding) features, and $C_i$ the observed confounders. Appendix E also lays out the key identification assumptions used throughout the method, including conditional exogeneity of the error term and an additive decomposition of confounder effects. Specifically, Appendix E formalizes one of our key assumptions: Exogeneity conditional on covariates, where $\mathbb{E}[\epsilon_i \mid X_i] = 0,$ where $X_i$ contains all observed variables. This assumption addresses your concern about whether the observed confounders are sufficient, as it states that once we condition on the available covariates, no residual dependence remains between the noise term and the predictors.
>
> (b) What DeconfoundLM does and how the confounder contribution is removed
>
> DeconfoundLM operates in two stages:
>
> 1. Estimate the confounder effect.
>    We estimate \(h(C_i)\) using an appropriate causal estimator (IV in our experiments; alternatives include Double Machine Learning or Adversarial GMMs).
>
> 2. Construct and use a deconfounded reward.
> We form
> $ \hat r_i = y_i - \hat h(C_i),$
>    and use $\hat r_i$ as the reward for policy optimization. This is the removal phase referred to in line 320.
>
> We will revise Section 4 so that this residual-based reward is explicitly introduced.
>
> (c) What happens after deconfounding
>
> After forming the deconfounded reward \(\hat r_i\), we fine-tune the policy with PPO. We currently mention this in Appendix D, but we will integrate the explanation into Section 4 and add an algorithm box summarizing the full pipeline.
>
> (d) Architecture: linear heads vs LoRA
>
> The paper uses two setups. In Section 3, we attach a linear head to the LLM embeddings and train it either with an MSE loss to predict CTRs in observational settings or a binary cross-entropy loss with pairwise experimental data. In Section 4, the policy model instead uses a LoRA adaptation of the base LLM, which we have mentioned in Appendix D. These details are already in the code repository of the project, and we will include these architectural and optimization details in Appendix D to make the setup fully reproducible.
>
>
> (e) Baselines
>
> We have the following Baselines:
>
> - Model with only sentiment / sentiment+noise: Reward modeling using the  corresponding target + PPO
> - RL w/ observed performance: Reward modeling using observational reward $y_i$ + PPO.
> - RL w/ popularity in text / in layer: Reward modeling using observational reward $y_i$ and popularity \(p_i\) is appended to text or included as a scalar feature + PPO
> - ODIN: implemented following Chen et al. (2024) to remove confounder effects.
>
> We will expand Appendix D to fully document these baselines.
>
> 2. **Limited novelty of Section 3** As you pointed out, our results in Section 3 demonstrate that classical confounding issues manifest directly in LLM fine-tuning and that model size interacts with confounding and regularization in non-trivial ways. This section is intended as motivation for the effect of confounding on LLM fine-tuning tasks rather than novelty. The main contribution of our paper is our proposed method that removes the effect of confounder-driven bias without relying on the counterfactual invariance assumption used in many reward misspecification literature. We will clarify this positioning in the introduction and related work.
>
> 3. **Evaluations** Thank you for pointing this out. Our choice of using semi-synthetic environment for evaluation was intentional, as it allows us to know the true data-generating process, ensuring that we can rigorously test whether various methods actually remove confounding influences. Using real-world data would require imposing additional modeling assumptions to infer the underlying structure, which can obscure whether improvements reflect correct deconfounding or mis-specification. Nonetheless, we agree the empirical section would benefit from more breadth. We will expand the experimental suite and include ablations to illustrate robustness.

---

> > ### Author Response · Authors · 2025-12-03
> > **Response to questions**
> >
> > 4. **Question: regularization scaling (Figure 5b)**
> >
> > We see how this could be confusing. Figure 5b only compares fixed $\lambda$ values up to 10,000, and within that subset $\lambda = 10{,}000$ appears best. Our statement about scaling regularisation comes from Figure 5a, which reports the optimal $\lambda$ for each model size. Aside from the smallest model, the optimal value increases with capacity. We will clarify this distinction in the revised text.
> >
> > 5. **Question: notation in Section 4 and confounder assumptions**
> >
> > In Section 4:
> >
> > - $X_i = (T_i, \tilde X_i)$ contains all observed features.
> > - $\tilde F_i$ is the subset of observed non-confounding features entering $g(\cdot)$.
> > - $C_i$ contains confounders affecting both text choice and the outcome.
> >
> > We treat $\tilde F_i$ and $C_i$ as disjoint sets, with the additive decomposition we have
> > $y_i = g(T_i,\tilde F_i) + h(C_i) + \epsilon_i.$
> >
> > As described above and in Appendix E, our key identification assumption is  $\mathbb{E}[\epsilon_i \mid X_i] = 0,$
> > which addresses your concern about whether the observed confounders are sufficient, as it states that once we condition on the available covariates, no residual dependence remains between the noise term and the predictors.
> > We will discuss these assumptions directly in Section 4 to clarify when the method applies and what conditions are required for identification.
> >
> > Thank you again for the detailed review. Addressing these issues, especially by adding missing methodological details, clarifying assumptions, and expanding the experimental section, will significantly strengthen the clarity of the paper, and we will incorporate these improvements in the revised manuscript.

---

### Official Review · Reviewer_2hgS · 2025-10-27

**Soundness:** 2
**Presentation:** 2
**Contribution:** 2
**Rating:** 2
**Confidence:** 3

**Summary:**

The paper tackles how to fine-tune language models on observational data (like clicks or engagement) without letting them overfit to spurious confounders such as timing or popularity. It proposes DeconfoundLM, a method that tries to isolate the causal effect of the model’s text on outcomes and train on that signal, and shows in controlled experiments that this can outperform standard RLHF/DPO-style approaches.

**Strengths:**

Importance of the problem: leveraging observational data is important, and taking into account confounders is even more crucial based on this context.

Empirical evidence for the problem tackled: the stackexchange and Upworthy experiments illustrate the potential of the LLM for learning spurious correlations from observational data. The Upworthy study also shows a nice insight: larger LMs overfit more strongly to confounded observational signals and require much heavier regularization.

**Weaknesses:**

Observed confounders: the method assumes access to known confounders and corrects for them, but the paper doesn’t clearly specify  how these confounders are encoded in practice with DeconfoundLM, or how robust the method is to missing/mismeasured confounders.

Violoation of the IV assumption: the method relies on an IV-style correction and assumes the IV only affects reward via popularity in the example in 4.1. However, the exclusion restriction is quite strong and deserved more discussion.

HPT tuning: regularization strength is chosen using experimental ground truth CTR, not purely observational data. This gives the observational models access to oracle feedback that wouldn’t exist in a purely observational setting.

Experimental setup: the method is validated using only one synthetic experiment, which is not sufficient. Furthermore, this synthetic setup is designed to satisfy the assumptions that the authors make (additive structure, exclusion,...) More experiments are required to illustrate the benefits of the method, for example training on real world observational data and evaluating on held-out A/B tests.

Lack of details: it is not clear how to plug the paper's method into RLHF/DPO pipelines. Should we deconfound before training the reward model, then do standard RLHF? Should we modify DPO-style objectives?

**Questions:**

See weaknesses section.

---

> ### Author Response · Authors · 2025-12-03
>
> We appreciate the reviewer’s thoughtful assessment of the paper and are glad that you found the problem setting important and the empirical demonstrations compelling. In particular, we are encouraged that you highlighted the relevance of studying observational data and confounding, and the empirical insights from the StackExchange and Upworthy experiments.
>
> Below, we address your comments individually and will incorporate all corresponding improvements in the updated manuscript.
>
> 1. **Use of observed confounders and robustness to missing ones**
>
> Thank you for raising this point. Equation 1 provides the structural decomposition we assume, where confounders enter the outcome through the term $h(C)$. DeconfoundLM’s objective is to estimate this component using tools such as IV estimation, Double Machine Learning, or Adversarial GMMs, and subtract it from the observed outcome before RL-based fine-tuning. Throughout the paper (Section 3.2 and Appendix E), we discuss that our method operates on observed confounders, similar as previous literature, but does not rely on the counterfactual invariance assumption that many prior works impose.
>
> While unobserved confounders may still exist, the ability to adjust for all available confounders substantially reduces the remaining bias, especially in industrial settings where rich logs and metadata are common. We will expand our discussion of this phenomenon and clarify how confounders are encoded in practice.
>
>
> 2. **Exclusion restriction and IV assumptions**
>
> Thank you for bringing this up. While identifying perfectly valid instruments can be challenging in many business applications, there is substantial prior work showing that suitable instruments do exist in practice. Imbens [1] provides an extensive econometric overview of IV methods, Sovey and Green [2] discuss their use in political science, and Wu et al. [3] offer a recent survey on IVs within causal inference and machine learning. In the Airbnb setting we reference, where price acts as a confounder for demand, supply-side factors can serve as plausible instruments [1]. We already acknowledge these requirements in Section 4, and we will make these discussions more explicit in the revised manuscript.
>
>
> 3. **Hyperparameter tuning in observational settings**
>
> In the Upworthy experiments, when models are trained on observational data, the regularization strength is selected solely using observational validation loss, not experimental CTR. As shown in Figures 4 and discussed in Section 3.2, relying only on observational validation can itself be problematic because the validation signals are also confounded. This is reflected in our results: the λ that minimizes validation MSE $(=18,000)$ differs substantially from the λ that yields the best test ROC AUC $(= 50,000)$. We will revise the manuscript to make this distinction more explicit and ensure that the tuning procedure cannot be misinterpreted as using oracle experimental feedback.
>
> 4. **Experimental validation and reliance on a single synthetic setup**
>
> Our choice of a controlled synthetic environment was intentional, as it allows us to know the true data-generating process, ensuring that we can rigorously test whether various methods actually remove confounding influences. Using real-world data would require imposing additional modeling assumptions to infer the underlying structure, which can obscure whether improvements reflect correct deconfounding or mis-specification. That said, we agree that additional settings, especially ones where assumptions such as additivity or exclusion do not fully hold, would strengthen the empirical section. We will expand the experimental suite and include ablations to illustrate robustness.
>
> 5. **Integration of DeconfoundLM with RLHF/DPO pipelines**
>
> Thank you for flagging the need for clarification here. As explained in Section 4, the aim of DeconfoundLM is to correct reward misspecification by purging the influence of confounders before the RL stage. After estimating and subtracting $h(C)$, we fine-tune the policy using standard RL. In our experiments, PPO is used for this step. This is mentioned in Appendix D, but we agree that the main text should make the pipeline clearer. We will add a more explicit description and diagram in the revised manuscript.
>
> We are grateful for your detailed feedback and thoughtful suggestions. Addressing these points will help us significantly improve the clarity, rigor, and completeness of the paper.
>
> References:
>
> [1] Imbens, G. (2014). Instrumental variables: an econometrician's perspective (No. w19983). National Bureau of Economic Research.
>
> [2] Sovey, A. J., & Green, D. P. (2011). Instrumental variables estimation in political science: A readers’ guide. American Journal of Political Science, 55(1), 188-200.
>
> [3] Wu, A., Kuang, K., Xiong, R., & Wu, F. (2025). Instrumental variables in causal inference and machine learning: A survey. ACM Computing Surveys, 57(11), 1-36.

---

### Official Review · Reviewer_FfU6 · 2025-10-29

**Soundness:** 2
**Presentation:** 2
**Contribution:** 2
**Rating:** 4
**Confidence:** 3

**Summary:**

The paper studies the risk of values of fine-tuning LLMs with historical observational data. It proposes a novel fine-tuning method, called DeconfoundLM, the debias the influence of observed confounders from the reward signals.

The paper provided well-controlled experiments to demonstrate the pitfalls and potentials  in SFT and DPO setting.

**Strengths:**

- The experimental design is clear and interesting to demonstrate the influence of latent confounders on fine-tuning.
- The proposed DeconfoundLM combined rigorous methods from causal inference to debias the influence from confounders.

**Weaknesses:**

- The paper is quite compact, and some important contents are not presented in the main body. This makes it difficult to have a detailed review. Please refer to the question part.

**Questions:**

- How does the proposed DeconfoundLM actually work? Readers may expecting a set of detailed equations with concrete examples. In addition, it would be much better to provide an algorithm box.
- What is the background behind the MIND dataset? and how to compare and interpret the *W*, *C*, and *E* metrics (are they defined?) in Table 1 and 2.

---

> ### Author Response · Authors · 2025-12-03
>
> We thank the reviewer for taking the time to evaluate our work and for highlighting several positive aspects of the paper. We appreciate your recognition that our experimental design illustrates how latent confounders can influence fine-tuning, and that DeconfoundLM combines rigorous causal-inference methodology with practical alignment concerns.
>
> Below, we respond to your concerns point-by-point and will incorporate all requested clarifications in the revised version.
>
> 1. **Paper compactness and missing details**
>
> Thank you for this observation. We organized the manuscript to keep the central empirical findings and conceptual contributions in the main body, which required placing extended derivations, dataset statistics, and additional details in the appendix. Based on your feedback, we will revise the paper to discuss more of these elements.
>
> 2. **Clarifying how DeconfoundLM works**
>
> We appreciate this suggestion. Section 4 and Equation 1 currently introduce the formal causal decomposition, and Section 4.1 provides the simulation functions (Equations 3–5) and explains why the IV estimator succeeds under our design. Nevertheless, an explicit operational description would substantially improve readability. Based on your feedback, we will improve the manuscript to make sure that we provide more details about the method and an algorithm box.
>
> 3. **Background on the MIND dataset and the meaning of W/C/E metrics**
>
> Thank you for the opportunity to clarify this. In our synthetic evaluations, we use the MIND dataset (Wu et al., 2020), which contains more than 160,000 news articles with headlines and full body text. We treat the article body as input and simulate performance scores using Equations 3 to 5. To maintain domain consistency, we restrict all experiments to its largest single topic: the sports category.
>
> Regarding the W, C, and E metrics: these represent the number of team mentions in generated headlines from:
> * W: West Coast teams
> * C: Central teams
> * E: East Coast teams
>
> These counts correspond directly to the popularity-based confounder defined in Equations 4 and 5. Because team-region popularity partially determines the simulated engagement scores, an increase in W/C/E mentions signals that the model is relying on the confounding popularity signal rather than the true causal signal (sentiment). This meaning is currently mentioned in the table title; however, we will revise the caption and include an explicit explanation in the main text to avoid any ambiguity.
>
> Thank you again for your thoughtful review and constructive feedback. We believe that addressing these points will significantly improve the clarity, completeness, and overall quality of the paper, and we look forward to incorporating your suggestions.

---

### Note · Authors · 2026-01-03

**Comment:**

We thank the review team for their careful reviews and constructive feedback. Based on the comments, we are working to improve clarity, better articulate the methodology and assumptions, broaden the experimental evaluation, and extend the methodology, which we believe will substantially strengthen the paper. We therefore withdraw this submission and plan to submit a revised version to another venue. We appreciate the reviewers’ time and insights.

**Withdrawal Confirmation:**

I have read and agree with the venue's withdrawal policy on behalf of myself and my co-authors.